# Lamiaceae Plants in Bulgarian Rural Livelihoods—Diversity, Utilization, and Traditional Knowledge

**Teodora Ivanova [1],\* , Yulia Bosseva [1], Mihail Chervenkov [1,2] and Dessislava Dimitrova [1],\***

1 Department of Plant and Fungal Diversity and Resources, Institute of Biodiversity and Ecosystem Research, Bulgarian Academy of Sciences, 1113 Sofia, Bulgaria; julibos@abv.bg (Y.B.); vdmchervenkov@abv.bg (M.C.)
2 Faculty of Veterinary Medicine, University of Forestry, 1797 Sofia, Bulgaria
\* Correspondence: tai@bio.bas.bg (T.I.); dessidim3010@gmail.com (D.D.)

**Abstract:** Lamiaceae comprises widely distributed medicinal and aromatic plants, many of which are traditionally used in European countries. The current study aimed to document Lamiaceae taxa used in rural Bulgaria (Southeast Europe) and to explore the related local knowledge and cultural practices that influence their utilization for various purposes. Field work included inventory of Lamiaceae diversity in home gardens and semi-structured interviews focused on the cultivation, collection, and utilization practices common among elderly inhabitants of 34 settlements in rural Bulgaria. We report the utilization of 27 Lamiaceae taxa, 9 of which were collected from the wild. Traditional and contemporary ways of utilizing Lamiaceae taxa as culinary and medicinal plants, in herbal teas, as repellents, ritual plants, etc., are presented. Recent knowledge on medicinal properties contributed to the introduction of new taxa in gardens (wild and cultivated), while traditional culinary practices were found to sustain the diversity of local forms (landraces).

**Keywords:** home gardens; culinary herbs; medicinal plants; ethnobotany; *Mentha*; *Satureja*





## 1. Introduction

Lamiaceae (Labiatae) is a cosmopolitan family and accounts for over 7000 species belonging to 245 genera [1]. Members of the family have long-lasting popularity for their diverse essential oil profiles, hence their multiple applications in medicine, cosmetics, gastronomy, etc. [2–4]. Many members of the Lamiaceae family used and cultivated since antiquity are still utilized [5–7]. Due to the broad spectrum of the biological activities (e.g., antimicrobial, antifungal, antioxidant, anti-inflammatory, anticancer, biocidal, etc.) of their secondary metabolites, Lamiaceae plants are used by local people based on their inherited empirical skills, or as a result of acquisition and exchange of traditional and/or modern knowledge [8–12]. Numerous ethnobotanical studies reported on the various past and current utilizations of different Lamiaceae taxa around the world and in Europe [4,13–21].

The status of Lamiaceae taxa as culinary and medicinal plants made them common in home gardens. Moreover, the growing urbanization and industrialization of agriculture turned several Lamiaceae taxa, e.g., *Lavandula*, *Ocimum*, *Origanum*, *Salvia*, etc., into important industrial crops that are grown around the world [22–24]. However, many of the traditionally used Lamiaceae plants are still harvested from the wild, often in an unsustainable way, which could be evaded through cultivation [25,26]. Global climate changes also threaten the crop yields and quality of production [26]. Large scale cultivation generally relies on limited biodiversity of crops which, in essential oil plantations, could result in a reduced range of bioactive constituents, flavors, and aromas, thus not meeting the expectations of different communities for their specific (traditional) medical practices, cuisines, and/or their cultural/spiritual needs. Conversely, small-scale farming offers opportunities to maintain a wider range of assorted crops, varieties, and/or landraces which not only provide for a more diverse diet and additional income for their owners,

but can also assist the preservation of valuable genetic resources, like valuable Lamiaceae plants [27–29]. In this sense, small farms and home gardens, as compact, diverse, and multilayered agroforestry systems provide for an important multitude of services to their owners and to local communities [30–33]. These environmentally friendly and more sustainable systems also cater to the conservation of wild and agrobiodiversity [34–37]. The owners of small farms and larger home gardens are more likely to use traditional agricultural practices and the related traditional knowledge inherited from previous generations, which is regarded as an important stepping stone for the implementation of agroecological principles in practice [38–42].

Local plant genetic resources preserved by gardeners and farmers, as well as the traditional knowledge related to their use and cultivation, however, are threatened by the gradual urbanization of the industrialized societies and growing depopulation and ageing of the inhabitants of the rural areas [43,44]. Plant diversity in European home gardens, and especially in those of Eastern and Southeastern Europe, remains understudied in comparison to home gardens and homesteads in the tropics, mostly due to the specific socio-economic impact of the latter [30–33]. Bulgarian rural home gardens, which currently range in size between several square meters and half a hectare, were found to provide substantially for the family sustenance, harboring a relatively large number of annual and perennial crop species [45]. Lamiaceae was found to be the second most represented plant family in Bulgarian rural home gardens, after Rosaceae, the latter being represented mainly by singular trees and shrubs cultivated for their fruits or grown as ornamentals [45]. On the contrary, in home gardens across Europe and the East Mediterranean, members of Lamiaceae were found more scarcely [46–48]. Additionally, it was found that most of the gardening area was cultivated on an annual basis, while herbs/spices, ornamentals, and fruit trees bordering the plots were more permanent elements.

The aim of the current study is to assess the variety of Lamiaceae taxa cultivated in Bulgarian home gardens as a function of their traditional and modern uses and to evaluate the factors/drivers that maintain and/or change their taxonomic diversity and related knowledge. We present the case in the frame of the local tradition to use and cultivate plants of the Lamiaceae family as medicinal and aromatic plants all over Bulgaria.

## 2. Materials and Methods

### 2.1. Study Area and Data Collection

Bulgaria is located in the center of the Balkan Peninsula, in the most southeast corner of Europe. The territory of the country is in the transitional area between the temperate and the Mediterranean climatic zones, with slightly elevated average annual temperatures and lower precipitation rates in the last three decades: hardiness zones 7–8 [49]. Bulgarian vascular flora comprises 4064 species of spermatophytes affiliated with 921 genera and 159 families, of which nearly 150 taxa belong to Lamiaceae [50,51].

Representatives of 74 households took part in the field study (2017–2022). They were from settlements in eight Bulgarian provinces (Blagoevgrad, Haskovo, Plovdiv, and Smolyan on the southern site of the Balkan Mt. range, and Lovech, Montana, Pleven, and Vratsa on the northern site, Figure 1).

Participants were recruited directly using the snowball sampling approach. Assistance of local leaders (mayors, local cultural activists, etc.) was acquired so as to identify prominent gardeners, agronomists, and/or local healers, when needed. Formal information on the age, education and occupation of every participant was collected. Informed consent was verbally obtained from every participant prior to the interview. The guidelines prescribed in the Code of Ethics of the International Society of Ethnobiology [52] were followed during the field study, and their compliance was confirmed by the Scientific Council of the Institute of Biodiversity and Ecosystem Research, Bulgarian Academy of Sciences, acting as independent institutional Ethics Board (Decision No. 6/21/05/21).

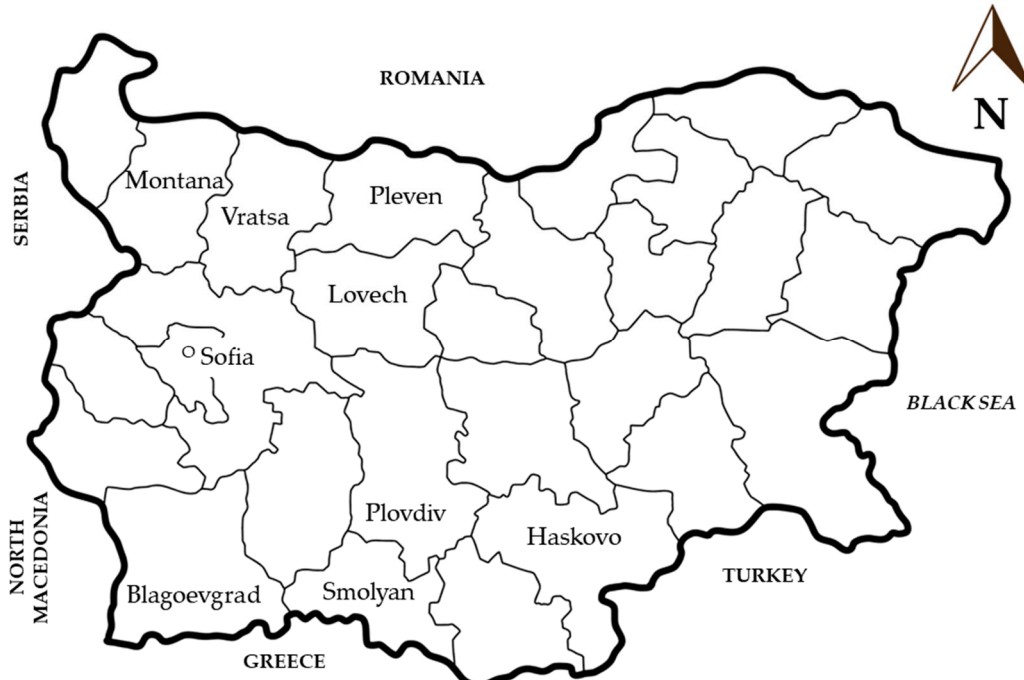

**Figure 1.** Map of the studied provinces.

Participants were asked to share information about the ways they currently utilize plants of the Lamiaceae family, as well as the taxa they cultivate in their home gardens and those collected from the wild. Information on the source of different crop plants was collected together with data on the processing of the collected/cultivated plants, if any. Participants were invited to freely list the plants they utilized. Additional information was asked if some plants were present in the garden or stored on a visible place in the living premises. Guiding questions on popular culinary herbs and herbal teas were asked, if necessary.

Voucher specimens and/or image data were collected for identification purposes; herbarium specimens were deposited in the Herbarium of the Institute of Biodiversity and Ecosystem Research, Bulgarian Academy of Sciences (SOM). The identification of the plants was carried out at least to the genus and species taxonomical levels in accordance with the Handbook of Bulgarian vascular flora [53]. Plant names are in accordance with the Plant list (2013) [1].

Presented plants/herbal products were summarized in eight use categories:

- Culinary herbs (CUL)—used fresh and/or cooked in preparation of salads and/or dishes;
- Ornamental plants (ORN)—grown for their ornamental flowers and/or foliage;
- Medicinal plants (MED)—used for healing purposes incl. for preparation of recreational herbal teas;
- Aromatic plants (AROM)—used for aromatization of the garden and/or of the home (fresh or dry);
- Insect repellents (REP)—for deterring biting and vexing insects (mosquitos, flies), agricultural pests and clothes moths;
- Pollinator attraction (POLL)—plants planted and/or reported to be grown around other crops so to attract pollinators;
- Symbolic plants (SYM)—fresh/dry used traditionally for decoration of the home and other buildings or for personal decoration following religious or other rituals;
- Technical plant (TECH)—used for making of the household and other objects.

Importance of each taxon was assessed using the use reports (UR) of the participants [54]. We visualized the multitude of utilization via Venn diagrams and compared the

use of the recorded taxa using the Jaccard (similarity) index (JI) for each pair of use records. Calculation of the JI was performed using the following formula:

$$JI\ (X,\ Y) = |X \cap Y|\ /\ |X \cup Y|,$$

where X and Y signify every two datasets. JI ranges from 0 (no similarity) to 1 (total equality) [55].

Sources of seeds and/or planting material were categorized as: local forms (landraces, purchased from the market/retail, introduced from the wild into the garden or collected from the wild for direct consumption and/or other uses).

### 2.2. Data Analysis and Statistics

Statistical association between nominal and ordinal variables was evaluated through chi-square tests (Fisher's exact test) and correlation analysis (Spearman rank-order correlation coefficient). All statistical tests were based on two-sided tests and with a significant level of at least $\alpha = 0.05$. Statistical analyses were performed using the SPSS statistical package (ver. 20.0, SPSS, IBM Inc., Armonk, NY, USA).

### 3. Results

Participants were primarily seniors, 90% over 56 years of age, actively engaged in home gardening (Table 1). The sets of single living participants and families were similar in number, 37 and 38, respectively. Education of the participants was prevalently secondary or higher. Half of the retired participants (79% of all participants) were professionally involved in industrial agriculture during their active years. However, even those who used to work or were still working in other professions were involved in agricultural activities throughout their lives. The retired family members were those responsible for decisions related to the composition of the gardens and spent most of their time in household organization and agricultural activities. Younger family members were those who were supplying new and foreign varieties for the gardens or helping in the processing of the garden yield, as well as introducing new recipes and knowledge.

**Table 1.** Characteristics of the participants.

| Characteristic | Total |
|---|---|
| Sex | |
| Single female; N (%) | 26 (34.7) |
| Single male; N (%) | 11 (14.7) |
| Family; N (%) | 38(50.7) |
| (N [1]) | 75 |
| Participant age (years) | |
| Median | 70.00 |
| Mean $\pm$ SD | 68.44 $\pm$ 10.19 |
| Age range (years) | 35–84 |
| 35–55 N (%) | 7 (11.3) |
| 56–75 N (%) | 40 (64.5) |
| 76–84 N (%) | 16 (25.8) |
| Education | |
| Primary; N (%) | 14 (18.7) |
| Secondary; N (%) | 34 (45.3) |
| College/University; N (%) | 19 (25.3) |

[1] Sample sizes vary due to missing data in the different variables. N: Sample size.

Most of the participants (69%) used two to four taxa of Lamiaceae, while those interested in six or more were relatively few (Figure 2). The number of those growing only one taxon was fairly small (three). These were usually people with very small gardens, keen in ornamental gardening with little or no interest in herbal teas. Age, sex, and education

were not significantly correlated with the number of cultivated/used taxa ($p > 0.2$). The highest number of taxa (nine) was recorded in a twin house with a common home garden managed by four people. There were no participants that would collect Lamiaceae plants only from the wild.

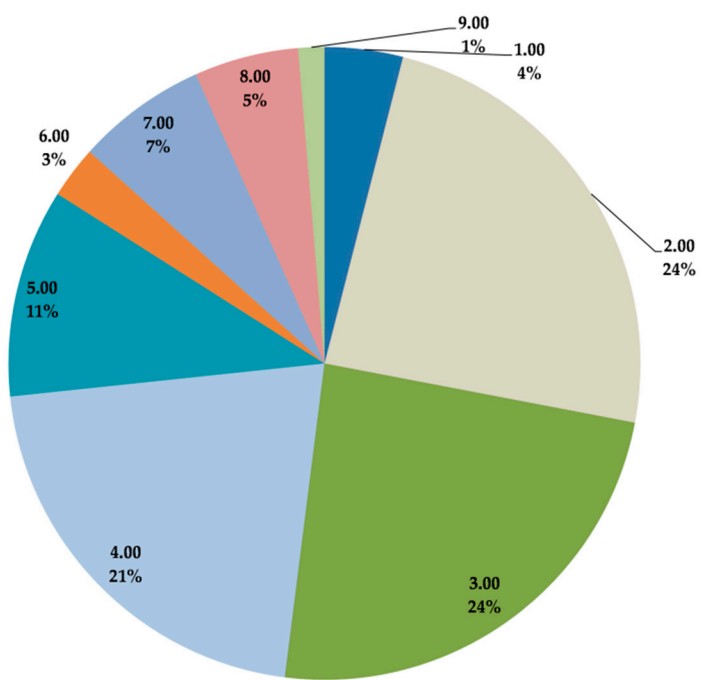

**Figure 2.** Number of used Lamiaceae taxa per participant (N, %).

Studied gardens were found to harbor small patches or separately planted individuals of 22 taxa of Lamiaceae, belonging to 15 genera (Table 2, Supplementary Table S1). Four of them were both cultivated and collected from the wild for personal/family use, while five species were collected by the participants only from the wild. *Salvia* was the most diverse genus with four cultivated species, followed by *Mentha* and *Satureja*, with three taxa each. The latter two genera concentrated most of the citations (76.5%). Third place was occupied by *Ocimum*, which was represented by two species, *Ocimum basilicum* L. and *O. minimum* L., the latter being rarely registered (28 and 4 mentions, respectively). *Clinopodium*, *Marrubium* and *Origanum* were presented by two taxa, one collected from the wild and one introduced into the gardens from the nearby populations and/or grown from seeds or planting material procured by informal exchange or from the market. Several participants demonstrated various *Thymus* sp. and *Mentha spicata* L. cultivars as well as *Origanum vulgare* subsp. *hirtum* (Link) Ietsw., imported from other European countries as culinary herbs. The remaining seven genera were represented by only one species.

**Table 2.** Cultivated and wild Lamiaceae taxa used in rural Bulgaria.

| Genus | Taxon/Voucher Specimen Collection Reference | Origin | Source | Occurrence, % of Gardens (Landraces [3]) | Provinces | UR | NU |
|---|---|---|---|---|---|---|---|
| *Agastache* | *Agastache foeniculum* (Pursh) Kuntze/BI300317_AF | I | C | 2.7 | Hs | 5 | 4 |
| *Clinopodium* | *Clinopodium dalmaticum* L./SOM177666; I040517_ClD | A | W | 0.0 | Sm | 2 | 1 |
| | *Clinopodium vulgare* L./B160617_ ClV | A | C, W | 1.3 | Pd | 2 | 1 |
| *Lamium* | *Lamium galeobdolon* (L.) Crantz/B140617_LG | I | C | 1.3 | Lv | 1 | 1 |

**Table 2.** *Cont.*

| Genus | Taxon/Voucher Specimen Collection Reference | Origin | Source | Occurrence, % of Gardens (Landraces [3]) | Provinces | UR | NU |
|---|---|---|---|---|---|---|---|
| *Lavandula* | *Lavandula angustifolia* Mill./B300317_LAn;B140617_LAn; | I | C | 13.3 | Bl, Hs, Lv, Mo, Pv | 19 | 4 |
| *Marrubium* | *Marrubium peregrinum* L./BI300817_MP | A | W | 0.0 | Hs | 1 | 1 |
| | *Marrubium vulgare* L./B140617_MV | A | C | 1.3 | Bl | 1 | 1 |
| *Melissa* | *Melissa officinalis* L./BI140617_MO; BI300317_MO; B140617_MO; BY300717_MO; B160418_MO | A | C | 24.0 | Bl, Hs, Lv, Mo, Pv | 26 | 4 |
| *Mentha* | *Mentha pulegium* L./B230419_MPu | A | W | 0.0 | Bl | 3 | 2 |
| | *Mentha spicata* L./B260719_Ms; B030821_MS; B040821_MS; B300317_MS; B040517_MS; B130617_MS; B140617_MS; B280617_MS; B300817_MS; B100719_MS; BI260819_MS | A, I | C | 98.7 (68) | All | 118 | 5 |
| | *Mentha × piperita* L./B300317_MxP; B040517_ MxP; B130617_ MxP; B140617_ MxP; B280617_ MxP; B300817_ MxP; B100719_ MxP; BI260819_ MxP | I | C | 26.7 | Bl, Hs, Lv, Mo, Pd, Pv, Sm | 35 | 3 |
| *Ocimum* | *Ocimum basilicum* L./SOM177657; SOM177659; B260819_OB; I101019_OB; | I | C | 37.3 (16) | All | 52 | 7 |
| | *Ocimum minimum* L./SOM177655; SOM177660; B050717_OM; I110917_OM; I120917_OM | I | C | 5.3 (1) | Hs, Sm, Vr | 7 | 3 |
| *Origanum* | *Origanum vulgare* L./B260819_OV; B280819_OV; B290819_OV, | A | W | 0.0 | Bl | 3 | 2 |
| | *Origanum vulgare* subsp. *hirtum* (Link) Ietsw./SOM177661; SOM177662 | A, I [1] | C, W | 21.3 | Hs, Lv, Pd, Pv, Sm | 25 | 5 |
| *Rosmarinus* | *Rosmarinus officinalis* L./B300317_RO; B010417_RO | I | C | 18.7 | Bl, Hs, Mo, Pd, Pv | 28 | 5 |
| *Salvia* | *Salvia aethiopis* L./B290819_SAE | A | C | 1.3 | Bl | 1 | 1 |
| | *Salvia officinalis* L./BI160617_SO; I300817_SO; B260819_SO; B030821_SO | I | C | 12.0 | Bl, Hs, Lv, Pd, Sm | 17 | 3 |
| | *Salvia splendens* Sellow ex J.A. Schultes/B130617_SSp | I | C | 1.3 | Lv | 1 | 1 |
| *Satureja* | *Salvia viridis* L./BI120917_SV | I | C | 1.3 | Sm | 1 | 1 |
| | *Satureja cuneifolia* Ten./SOM177658 | A | W | 0.0 | Bl | 1 | 1 |
| | *Satureja hortensis* L./SOM177656; BI150617_SH; BI260819_SH; I020821_SH; B030821_SH | I | C | 74.7 (55) | All | 57 | 2 |
| | *Satureja pilosa* Velen./SOM177663 (c); SOM177664 (w); SOM177665 (w) | A | C, W | 1.3 | Hs | 3 | 1 |
| *Sideritis* | *Sideritis scardica* Gris./B310317_ SSc; B310317_SSc | A [2] | C | 5.3 | Hs | 7 | 2 |
| *Stachys* | *Stachys byzantina* K. Koch/B140617_StB | I | C | 2.7 | Bl, Mo | 2 | 1 |
| *Thymus* | *Thymus* sp./I040517_Th | A, I | C, W | 2.7 | Bl, Sm | 5 | 3 |
| | *Thymus vulgaris* L./B040517_TV; B280617_TV; B100719_TV; B030821_TV | I | C | 5.3 | Pd, Pv, Sm, Vr | 6 | 3 |

[1] Cultivated plants were introduced from nearby wild populations or were grown from seeds/planting material procured from elsewhere; [2] cultivated plants were grown outside the natural distribution of the species; [3] number of local forms (landraces) if any; A: autochthonous; I: introduced; C: cultivated; W: wild; UR: use-reports; NU: number of uses; provinces (Bl: Blagoevgrad; Hs: Haskovo; Lv: Lovech; Mo: Montana; Pd: Plovdiv; Pv: Pleven; Sm: Smolyan; and Vr: Vratsa).

Altogether, ten taxa were accountable for the 88.1% of all mentions for utilization of Lamiaceae plants led by *Mentha spicata* L. and *Satureja hortensis* L. They also encompassed most of the local forms (landraces) that participants were maintaining in their gardens, 68 and 55, respectively. *Ocimum basilicum* and *O. minimum* were the other taxa of which local forms were preferred.

Each participant cited no more than four uses per taxon, which was related not only to the overall use of the taxon, but also to differences in utilization of varieties/local forms. *Ocimum basilicum* was the species with the highest number of uses (seven uses). Additionally, *O. basilicum* was, respectively, mentioned two and three times less frequently than the most popular—*M. spicata* and *S. hortensis*. More than half of the taxa (55.5%) had one or two uses of which 11 taxa had only one use. Culinary use was about twice as popular as the medicinal one.

Lamiaceae plants were used most often (61.4% of the use-reports) as culinary herbs, followed by those appreciated for their ornamental value (39.3%). The similarity between CUL and MED categories on a taxonomical level was high (JI = 0.83) (Figure 3). However, the varieties/local forms demonstrated as culinary herbs were rarely appreciated for other purposes, with negative correlation being significant (Spearman rank correlation *p* < 0.01, Table 3) for four out of seven categories (ORN, MED, AROM, and SYM).

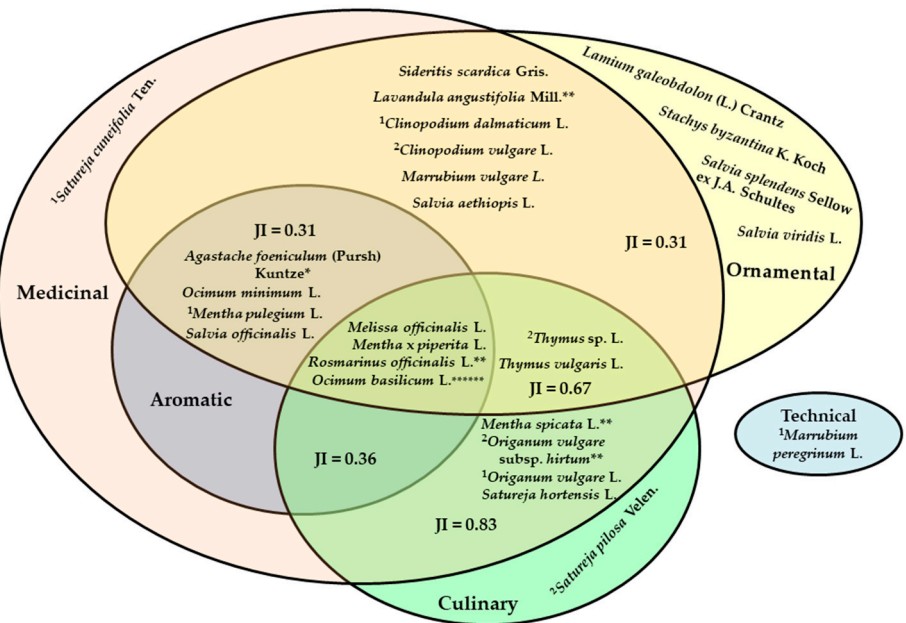

**Figure 3.** Multipurpose utilization of Lamiaceae representatives in rural Bulgaria. [1] Taxon collected only from the wild; [2] Taxon cultivated and collected from the wild; pollinator attraction; * repellent; ****** are the three pollinator attraction; * repellent; ** (*Ocimum basilicum* L. only) symbolic together; JI: Jaccard similarity index.

For example, *O. basilicum* was more frequently mentioned as an ornamental and aromatic plant than as a culinary one for which mostly foreign broad-leaf varieties were considered. On the other hand, only locally grown white flowering forms with small leaves were used for ritual purposes, but they were not consumed (Figure 4). *Ocimum basilicum* was the only Lamiaceae representative used symbolically in different rituals and for its protective (apotropaic) powers. Flowering *O. basilicum* branches were used by Ortodox Christians as church and home decoration and in various ceremonies as well as in rituals related to birth, weddings, and burials.

**Table 3.** Usage of Lamiaceae plants in rural Bulgaria and correlation between different usages.

| Use | CUL [1] | MED | AROM | REP | POLL | TECH | SYM | Total Number of Use-Reports | Total Number of Taxa |
|---|---|---|---|---|---|---|---|---|---|
| ORN | −0.307 ** | 0.081 | 0.169 ** | 0.100 | 0.018 | −0.048 | 0.082 | 112 | 17 |
| CUL | | −0.562 ** | −0.331 ** | −0.103 | −0.106 | −0.075 | −0.185 ** | 175 | 11 |
| MED | | | −0.03 | 0.003 | 0.036 | −0.039 | −0.043 | 86 | 21 |
| AROM | | | | −0.029 | 0.099 | −0.022 | 0.172 ** | 34 | 10 |
| REP | | | | | 0.183 ** | −0.013 | 0.085 | 13 | 5 |
| POLL | | | | | | −0.005 | −0.012 | 2 | 2 |
| TECH | | | | | | | −0.009 | 1 | 1 |
| SYM | | | | | | | | 6 | 1 |

[1] Abbreviations—CUL: culinary herb: ORN: ornamental plant; MED: medicinal plant, incl. herbal teas; AROM: aromatic plant; REP: insect repellent; POLL: pollinator attraction; SYM: symbolic plant used ritually; TECH: technical plant (brooms). ** Spearman correlation coefficient is significant at the 0.01 level (2-tailed).

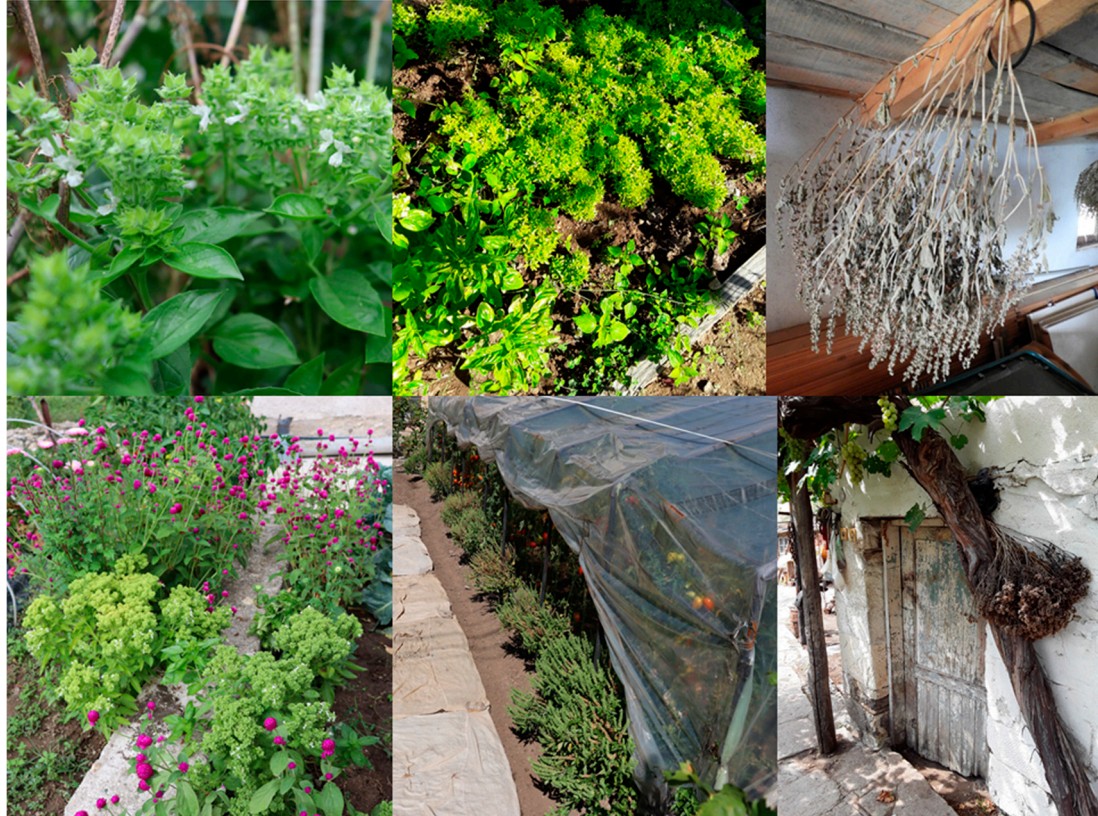

**Figure 4.** *Ocimum basilicum* L. in Bulgarian rural livelihoods.

Top (left to right): local landrace 'Zhenski bosilek' (women's basil) used as ritual and medicinal plant in Banichan village; *Ocimum basilicum* L. and *O. minimum* L. from retail seeds in Central Rhodope Mts.; and the drying of flowering whole plants of *O. basilicum.* Bottom (left to right): ornamental patch of *O. basilicum* and *Gomphrena globosa* L.; *O. basilicum* sown in front of greenhouse to attract pollinators; and the ritual decoration of *O. basilicum* for home protection.

Half of the studied taxa (11) were reported to be edible with only one collected exclusively from the wild (*Origanum vulgare* L.). The latter was also cited as spirit flavouring and as a medicinal plant. *Mentha spicata*, *S. hortensis*, and *O. vulgare* subsp. *hirtum* were most frequently used for preparation of traditional dishes. All culinary herbs were used dried, and only *M. spicata* (Figure 5) was also traditionally used fresh in the preparation of lamb meat, bean, and vegetable stews. Dried herbs were stocked in bunches or sealed

in containers and bags in quantities matching the usual demands of the household until the next harvest. Fresh herbage or leaves of other taxa were used only for preparations of foreign dishes like *O. basilicum* for Italian dishes and salads, *Mentha* × *piperita* L. and *Melissa officinalis* L. for desserts and cocktails, as well as *Thymus* sp. and *Rosmarinus officinalis* L. for roasted meats. Interestingly, the latter was appreciated for its aromatic and decorative foliage rather than as a culinary herb, and some of our respondents shared that they are aware of its culinary use but never used it in their cooking.

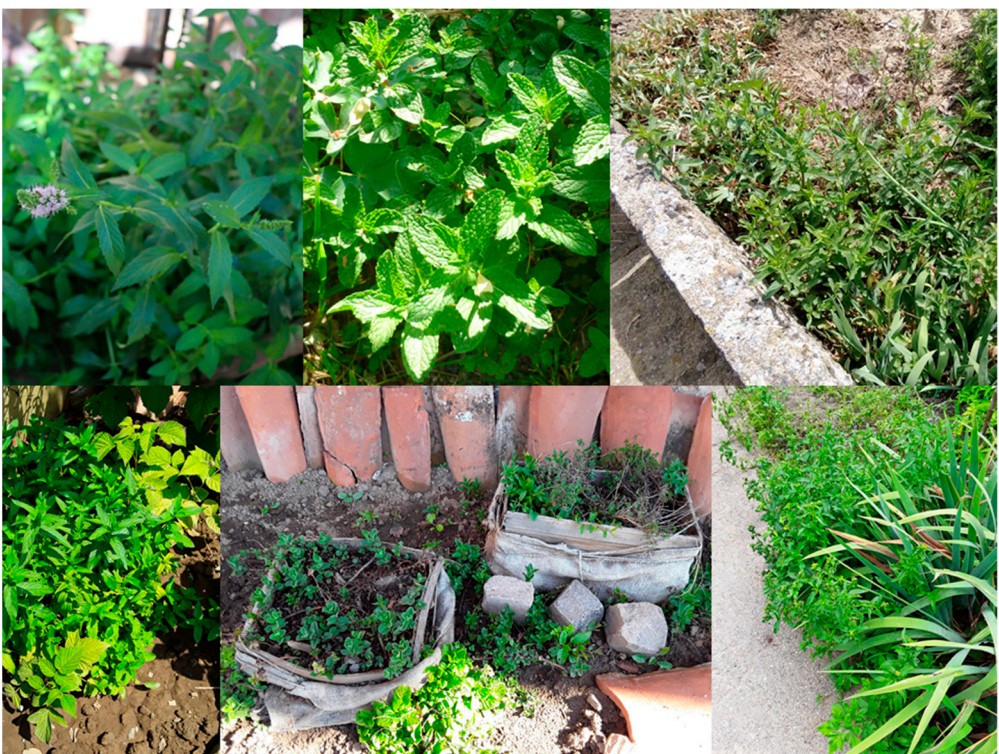

**Figure 5.** Variety of *Mentha spicata* L. local forms (landraces) in Bulgarian rural home gardens.

Twenty-one taxa were used for their healing properties, with *M.* × *piperita*, *M. officinalis*, and *Salvia officinalis* being the most commonly mentioned. Still, only 30.2% of the reports were related to the healing properties of Lamiaceae taxa. For medicinal purposes, dried plants were used as infusions. Some of the participants consumed them daily as healthy herbal teas, while others associate herbal teas with childhood illnesses and reject to cultivate and/or collect medicinal plants. Taxa used for household aromatization (10 taxa), insect repellents (5 taxa), and as pollinator attraction/honey plants (2 taxa) overlapped completely with medicinal plant category. Dried *Lavandula angustifolia* Mill. and *Mentha spicata* bunches were reported as effective in the repelling of clothes moths. *Origanum vulgare* subsp. *hirtum* and *M. spicata* were demonstrated as pantry pest repellents, placed directly in the containers with legumes, grains, etc. *Ocimum balisicum* and *Rosmarinus officinalis* were grown in the gardens next to windows to repel mosquitoes.

Only one species was mentioned as a technical plant; *Marrubium peregrinum* L., which was used for garden brooms that were made of large, sturdy herbage collected towards the end of vegetation. (Figure 6). However, the local knowledge on broom making was fading away, demonstrated by only one participant. On the other hand, the use of Lamiaceae plants to attract pollinators in greenhouses so as to promote tomato pollination should be regarded as a relatively new practice, popular among gardeners interested in eco-friendly agriculture.

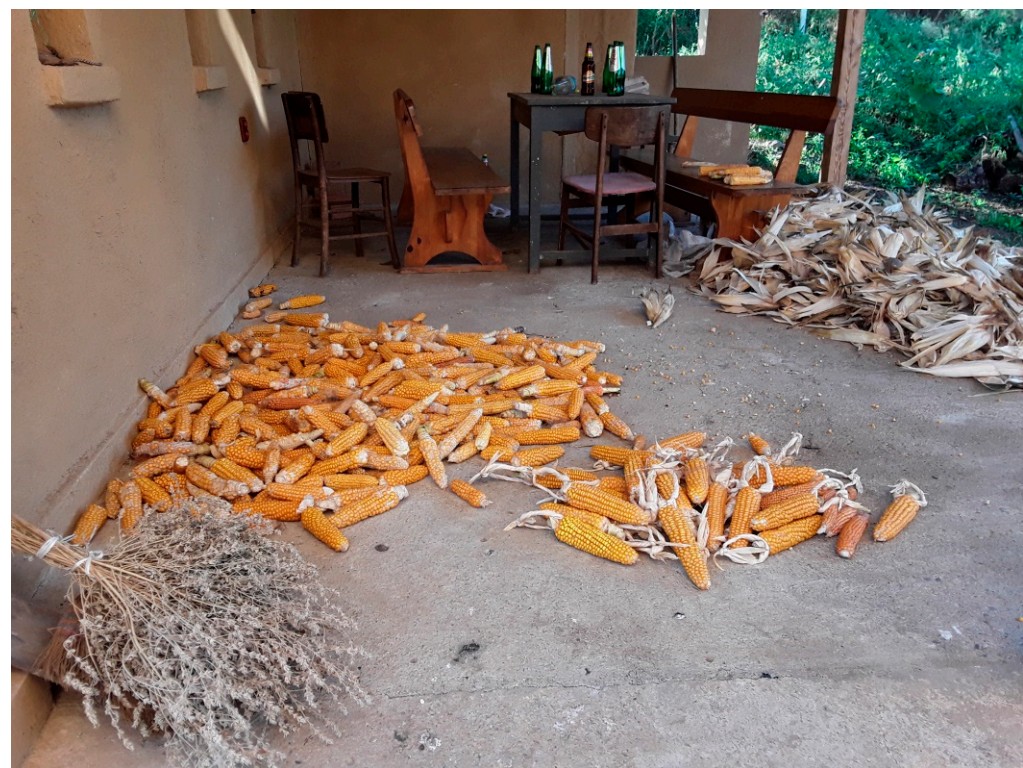

**Figure 6.** Home yard broom made of *Marrubium peregrinum* L.

The culinary use of some species resulted in the preservation of numerous local forms (Table 4). Contrastingly to the ornamental, aromatic and even medicinal plants that were sourced mainly from the market, seeds, rhizomes, and/or plantlets for culinary herbs were predominantly personally propagated. These were handed down through the generations and were replaced through exchange within the community only if inherited plants were lost. The practice of introducing plants from wild populations into home gardens was most popular for medicinal plants. Among these taxa, *Sideritis scardica* Gris., a species that became popular for its health benefits in recent years, was the only one that was transferred outside the area of its natural distribution. Cultivation of Lamiaceae species was preferred not only for the convenience and availability, but also due to the popularity of the protected status of some of the medicinal plants (e.g., *O. vulgare* ssp. *hirtum*, *S. scardica*). Additionally, our participants were more inclined to introduce wild-growing culinary herbs into their gardens than to lose time visiting natural populations.

**Table 4.** Sources of Lamiaceae taxa used in rural Bulgaria.

| Source | ORN [1] | CUL | MED | AROM | REP | POLL | TECH | SYM |
|---|---|---|---|---|---|---|---|---|
| Local forms [2] | 40 | 126 | 8 | 10 | 4 | 0 | 0 | 6 |
| Introduced from the wild | 6 | 7 | 20 | 3 | 1 | 0 | 0 | 0 |
| Retail | 51 | 29 | 38 | 13 | 5 | 2 | 0 | 0 |
| Informal exchange | 13 | 5 | 8 | 7 | 1 | 0 | 0 | 0 |
| Collected from the wild [3] | 0 | 5 | 9 | 1 | 0 | 0 | 1 | 0 |
| Fisher exact test, significance ($p$) | <0.001 | <0.001 | <0.001 | 0.008 | 0.591 | 0.374 | 0.06 | 0.310 |

[1] Abbreviations—CUL: culinary herb: ORN: ornamental plant; MED: medicinal plant, incl. herbal teas; AROM: aromatic plant; REP: insect repellent; POLL: pollinator attraction; SYM: symbolic plant used ritually; TECH: technical plant (brooms); [2] data are presented as a number of reports; [3] plants collected regularly from the wild for direct consumption or other use.

Additionally to cultivated Lamiaceae taxa studied gardens harbored several wild Lamiaceae species (*Ballota nigra* L., *Lamium amplexicaule* L., *L. maculatum* L., *L. purpureum* L., *Leonurus cardiaca* L., *Glechoma hederacea* L., *Salvia nemorosa* L., *S. pratensis* L. *S. verticillata* L.) that were considered by the participants as weeds without practical application.

## 4. Discussion

The overall number of Lamiaceae family members (22) found cultivated in Bulgarian rural home gardens was similar to that reported for other European home gardens, however, none of the taxa mentioned in these studies were present even in half of the studied gardens [46,56]. In the current study, two species, namely *M. spicata* and *S. hortensis*, were almost compulsorily present in the home hardens, grown in 98% and 74% of the studied gardens, respectively. Although Lamiaceae taxa were grown in limited quantities in Bulgarian home gardens, it is noteworthy to mention that their occurrence was comparable with many of the garden vegetables that were found to be main crops in cultivation [45,57]. Even some Lamiaceae taxa, which are not so popular as culinary herbs in Bulgaria (e.g., *O. basilicum*, *R. officinalis*), were also found more frequently cultivated in the Bulgarian home gardens than in other European countries [29,48,58,59]. The observed high diversity of Lamiaceae taxa in Bulgarian rural home gardens should be attributed not only to the fact that nearly 70% of our participants cultivated two to four Lamiaceae taxa in their gardens, but also to the very high preference for local forms (landraces) of culinary herbs that were inherited from previous generations or obtained through informal exchange within their community. Utilization of local genetic resources for their local organoleptic perceptions and their perceived cultural value was previously shown to contribute to their in situ preservation, which creates additional opportunities for the development of local entrepreneurship [60,61]. Still, the relatively high age of the rural population in Bulgaria should be considered as an alarming factor in terms of the need for the development of targeted collection programs for the safeguarding of these resources [62].

*Mentha spicata* and *S. hortensis,* the culinary herbs that occupied the first two places among the reported taxa, also had the highest number of local forms. On the other hand, other, incl. wild-growing members of Lamiaceae, such as *Origanum*, *Thymus,* and *Mentha*, popularly used in different dishes, preserved and fermented foods in the Mediterranean area were far less consumed by Bulgarians [63–66]. Specific plant spices and their combinations play key roles in the local cuisines that underline their uniqueness [67–69]. Practicing of traditional (agro)ecological knowledge is of crucial importance, as demographic and socioeconomic changes in the rural areas, especially in industrialized societies, gradually diminish natural human connectedness [4,70,71]. While utilization of *M. spicata* was reported from all over the world [29,72–74], *S. hortensis* was found popular both as a culinary herb and a medicinal plant mainly in the Balkans, Iran, and Turkey [75–77]. In this sense traditional culinary use underpins the maintenance of local Lamiaceae diversity. Recent results from the Adriatic area show that traditional knowledge related to usage of certain species is not only diminishing, but also changes in agricultural practices and land use are making home gardens the main source of medicinal plants, rather than the wild [78]. Still, similarly to other studies, it is hard to designate every inherited form with a landrace status, as no genetic and/or phytochemical analyses were performed during the study [56]. Nonetheless, the high number of gardens in which *M. spicata* and *S. hortensis* were recorded highlights the importance of home gardens for the preservation of local plant genetic resources, and also urges for their more detailed characterization, given the numerous factors that cause genetic erosion and loss of crop diversity [59,79].

Traditionally, consumed plants that have two or more other applications and are relatively easy to grow would possibly attract more attention even from unexperienced gardeners, which would contribute to the broadening of the impact of home gardens for the safeguarding of local plant diversity [80–82]. In the current study, single use was mostly a signifier for recent introduction of taxa into the gardens (e.g., ornamental varieties) or of outdated practices such as the preparation of brooms from the herbage of *Marrubium*

*peregrinum*. Others, such as *Clinopodium dalmaticum* L., have a limited distribution range in the country, which could be regarded as a restrictive factor for its usability [83]. Here, it is important to distinguish the *O. basilicum*, which was the only species with symbolic (ritual) use. Sweet basil is one of the most important ritual plants in Bulgarian Orthodox Christian traditions [84], and we found the local forms used for ritual purposes to be rarely consumed fresh or cooked. Similarly, *O. minimum* was perceived mostly as an aromatic and ornamental plant. The discerning and preservation of landraces and varieties of both *O. basilicum* and *O. minimum*, however, would be a complicated enterprise due to the high variability in genome size and chromosome number, plant morphology, essential oil profile, etc., that have created considerable taxonomical ambiguities throughout the years and were related also to the frequent intrageneric hybridization and extensive breeding of *Ocimum* [85,86].

While almost all of the reported taxa were known as medicinal plants, the number of use reports related to medicinal properties was less than 1/3. The number of taxa used in this category (21) was about twice as low when compared to ethnographical data previously reported for Bulgaria, however close to the number of Lamiaceae taxa used in other parts of the Balkans as medicinal plants [77,87,88]. Given the fact that in Bulgarian folk medicine several species of one taxa could be used for various ailments, sometimes interchangeably (e.g., *Lamium album* L., *L. maculatum*, *L. purpureum*), it is probable that knowledge of the uses of other members of Lamiaceae have remained preserved outside the assessed areas/among other communities [77]. Nonetheless, we should consider the transformative role of the publication of numerous phytotherapy books that provided the Bulgarian public with modern and/or foreign knowledge on the use of indigenous and introduced plant taxa since the 1950s [89–92]. Many of these books also provided information on the threats to natural populations of some medicinal plants and recommend more responsible utilization of the natural resources. The latter could also explain the low number of wild medicinal plants of the Lamiaceae family (8) used traditionally by Bulgarian farmers in the early 1990s [93]. Additionally, only five species of the Lamiaceae family, all of which were also shown here, were reported for recreational teas used in Bulgaria [94]. It is important to mention that Bulgarian folk medicine was, and in some places still is, typically practiced by skilled healers (called *bilkari*, *bilyari*, *znahari*, etc.) who collect, supply and often process medicinal plants that are further used by the patients. Hence, a substantial part of Bulgarian traditional knowledge related to medicinal plants was never largely available [77]. This could explain why the plants mentioned by our participants as medicinal were mainly sourced from the market and were exclusively consumed as infusions, most of which are taken as daily herbal teas. Common examples for such plants are *Thymus* spp., *Origanum vulgare*, and *Sideritis scardica*, which are popular winter teas sold allover Bulgaria. However, the traditional knowledge related to their use would be hard to trace. Complementary to the initial interest in the medicinal properties, we observed that the purchased plants were often appreciated as ornamental and aromatic plants. Still, utilization of Lamiaceae taxa for their healing properties was found to be an important factor for plant domestications in gardens which was found to be a useful approach to alleviate pressure on the natural populations of medicinal plants [33,95,96].

In conclusion, the cultivation of members of the Lamiaceae family in Bulgarian home gardens accommodates both the various needs and interests of their owners, as well as the preservation of wild and agrodiversity. While the medicinal properties of these taxa cater to higher diversity on a species level, local culinary practices were found to sustain the variety of local forms (landraces) that underline the role of home gardens as important pools of plant genetic resources that should be preserved and further explored in the frame of the multitude of benefits provided by these plants.

**Supplementary Materials:** The following supporting information can be downloaded at: https://www.mdpi.com/article/10.3390/agronomy12071631/s1; Table S1: Use reports and sources of the Lamiaceae taxa used in rural Bulgaria.

**Author Contributions:** Conceptualization, T.I. and D.D.; methodology, T.I. and D.D.; formal analysis, Y.B.; investigation, T.I., Y.B., M.C. and D.D.; data curation, Y.B.; writing—original draft preparation, T.I.; writing—review and editing, T.I., Y.B., M.C. and D.D. All authors have read and agreed to the published version of the manuscript.

**Funding:** The study is funded by the Bulgarian Ministry of Education and Science under the National Research Programme "Healthy Foods for a Strong Bio-Economy and Quality of Life" approved by DCM # 577/17.08.2018. Part of the field studies were supported under the project DN10/1/2016 "The Garden: Site of Biocultural Diversity and Interdisciplinary Junction" funded by the National Science Fund.

**Institutional Review Board Statement:** The study ethical compliance was confirmed by the Scientific Council of the Institute of Biodiversity and Ecosystem Research, Bulgarian Academy of Sciences, acting as independent institutional Ethics Board (Decision No. 6/21/05/21).

**Informed Consent Statement:** Informed consent was obtained from all participants involved in the study prior the interviews.

**Acknowledgments:** Authors are grateful to all participants for the shared knowledge and kind cooperation during the field study.

**Conflicts of Interest:** The authors declare no conflict of interest.

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
