# Peer review of "Lamiaceae Plants in Bulgarian Rural Livelihoods—Diversity, Utilization, and Traditional Knowledge"

_agronomy, doi:10.3390/agronomy12071631_

Round 1

Reviewer 1 Report

The paper “Lamiaceae plants in Bulgarian rural livelihoods – diversity,  utilization and traditional knowledge” evaluates Lamiaceae taxa cultivated  in Bulgarian home gardens as a function of their traditional and modern uses. The topic of paper is interesting and in the aim of the special issue of the journal. Congratulations to the authors!!

In my opinion the paper can be published  in present form.

Author Response

Dear Reviewer,

Thank you for the positive review and for the careful consideration to our work. We tried our best to improve the quality and readability of the current manuscript integrating the suggestions of all reviewers. Please, find the revised text and figures in the provided file.

Sincerely,

The Authors

Reviewer 2 Report

Dear Editor, dear Authors,

I've completed my review of the manuscript entitled Lamiaceae plants in Bulgarian rural livelihoods – diversity, utilization and traditional knowledge, authors Teodora Ivanova * , Yulia Bosseva , Mihail Chervenkov , Dessislava Dimitrova 

The authors present original results of their terrain work; in focus is etnobotanical survey of Lamiaceae taxa used in rural Bulgaria, aimed to explore the related local knowledge that influence their utilization for various purposes. The owners of small farms are more likely to use traditional agricultural practices and the related traditional knowledge inherited from previous generations which is regarded as an important stepping stone for the implementation of agroecological principles in practise, that's why this thematic is worth to explore in detail.

In this study, authors compiled a dataset of 74 households, during 2017–2022 growing season. They covered quite larg research area (eight Bulgarian provinces). Tipically ethnobotany survey methods (snowball sampling approach) were used and authors performed all required activities beside (followed  Code of Ethics of the International Society of Ethno-biology, and informed consent). Herbarium specimens were deposited in the official Herbarium.

In section 2.1. Study area and data collection  I would recommended finishing Figure 1 (particular comments in pdf).  Given that the area under study is on the border with other countries, questions for authors: have you perhaps researched the impact of human migration (intermarriage) and the consequent change in habits and traditions?

The results showed no much taxa (27 Lamiaceae taxa), but objectively described the stage of already started errodification of traditional knowledge. Each participant cited no more than four uses per taxon which was related not only to the overall use of the taxon but also to differences in utilization of varieties/local forms.

In discussion authors  discuss about Balcan and Mediterranean area, here I recommended some particular research and papers from that area (Adriatic Croatia and Slovenia) and thematic (medicinal Lamiaceae, influence of the state borders on traditional knowledge) that would be good to compare. Links:

https://www.mdpi.com/2223-7747/10/10/2087

https://ethnobiomed.biomedcentral.com/articles/10.1186/s13002-019-0332-1

https://www.frontiersin.org/articles/10.3389/fphar.2021.623070/full

On the other hand, the authors critically evaluated other similar researches (rural and urban gardens) and conclude that the cultivation of members of Lamiaceae family in Bulgarian home gardens accommodates both various needs and interests of their owners and the preservation of wild and agrodiversity.

Discussion is very well done, I congratulate to authors and I don’t have other comments on this part. I find the presented results here very important, because today we live in a society that is more aware of the need to conserve habitats and taxa biodiversity, especially ethnobotanic traditional knowledge which is important for preserving the identity of people! Therefore, such background information should be published to become accessible for broader audience.
The paper is written very well; results are suitable for the publication in  Agronomy. I marked these comments and suggestion in the attached file.

I herewith recommend the reviewed manuscript for publication in the Agronomy publons.

Author Response

Dear Reviewer,

Thank you for the recommendations and for the careful review to our work. We are grateful for the detailed assessment that will improve the quality and readability of the current manuscript. Please, find bellow the responses to the specific comments.

In section 2.1. Study area and data collection  I would recommended finishing Figure 1 (particular comments in pdf). Given that the area under study is on the border with other countries, questions for authors: have you perhaps researched the impact of human migration (intermarriage) and the consequent change in habits and traditions?

Thank you for this suggestion. We corrected the provided map in the manuscript. Effects of migrations (historical and recent) were spotted during our interviews with some of our participants, mostly from Haskovo province where big population of Anatolian Bulgarians (historical diaspora of Orthodox Christians from Aegean Thrace and Asia Minor moved back to Bulgaria in the first decades of XX century) resides and on the other hand part of the Muslim communities left the country in the recent 40 years. Although it was not related to any of the Lamiaceae species, we recorded some specificities related to wild food plants part of which could be found published here  https://www.webofscience.com/wos/alldb/full-record/BCI:BCI201900239954. We continue our work in these areas as well.

The results showed no much taxa (27 Lamiaceae taxa), but objectively described the stage of already started errodification of traditional knowledge. Each participant cited no more than four uses per taxon which was related not only to the overall use of the taxon but also to differences in utilization of varieties/local forms.

Thank you for the comment. Given the strong focus on food production in Bulgarian rural gardens (https://www.mdpi.com/2223-7747/10/11/2520/htm) and the high rates of aging population it would be interesting to follow the changes in utilization, especially in relation to new knowledge on foraging and botanicals used as medicines, cosmetics, etc.

In discussion authors discuss about Balcan and Mediterranean area, here I recommended some particular research and papers from that area (Adriatic Croatia and Slovenia) and thematic (medicinal Lamiaceae, influence of the state borders on traditional knowledge) that would be good to compare. Links: https://www.mdpi.com/2223-7747/10/10/2087

https://ethnobiomed.biomedcentral.com/articles/10.1186/s13002-019-0332-1

https://www.frontiersin.org/articles/10.3389/fphar.2021.623070/full

Thank you for these suggestions and provided current data. We elaborated on them in the Discussion (see line 344 onward).

Sincerely,

The Authors

This manuscript is a resubmission of an earlier submission. The following is a list of the peer review reports and author responses from that submission.

Round 1

Reviewer 1 Report

Review Agronomy

The MS is generally well presented and written. The weak part is the used methods, and how authors show their results. In my view, the MS includes a very small amount of Lamiaceae species, considering the high biodiversity of wild ones (and more considering the variety of cultivated ones). Also, the interview effort seems to be weak (only 75 interviews for 29 taxa).

Results show a low number of repetition of uses, and the extensive use of indexes is not justified, since the dataset is not big, and the somehow hide the low number of UR and studied taxa (or informants). What is the sense of describing in the table the RFC when it is just the UR/79?

Weak points

abstract

1.      As the paper dont deal woith chemical compounds or pharmacological properties, the sentence in abstract “ However, the extensive pharmacological research 12 throughout the years and universal commercialization of essential oils and extracts based 13 medicines derived from Lamiaceae plants have led to considerable replacement of the tradi-14 tional knowledge related to their use.” Is out of context

2.       “Current study aimed to document Lamiaceae taxa used…” / “Traditional and contemporary ways of utilization of Lamiaceae taxa as culinary 21 and medicinal plants, in herbal teas, as repellents, ritual plants, etc. are presented” in fact, as neither the table of results nor the sup. Mat. Deep on the real traditional use, describing the real use (use, medicinal use, condition, part used, administration / elaboration forms….) I would change this sentence.

Methods

1.      Is the study covering all Bulgarian territory? At least all provinces or all the biogeographical different units?

2.      A map on the surveyed population would be appreciated, to know the extension of the study and how it can reflect the real situation in the country.

3.      Lamiaceae has 150 wild taxa in the Bulgarian flora. Is this paper with just 9 of them included, well designed?

4.      Selection of informants. Ethnobotany use to deal with oral transmitted knowledge, as it study the folk knowledge, appart from the academic one. Then… your selection of informants is not good, as they were “was acquired so to identify prominent gardeners, agronomists and/or local healers, when needed.….” This is not ethnobotany.

5.      “Reference specimens and/or image data were collected for identification pur-112 poses; herbarium specimens were deposited in the Herbarium of the Institute of Bio-113 diversity and Ecosystem Research, Bulgarian Academy of Sciences (SOM).” Then, were are the voucher codes? They should be always added in the research (as mandatory in most journals dealing with the discipline)

6.      Taxonomy and nomenclature. 1. Please make sure authorities are always added at least the first time each taxa is mentioned in the text (see, e.g., ). Some odf the used names, even being in consonance of ref 51, are not generally considered valid at the used range: e.g., Ocimum minimum (see, eg. https://powo.science.kew.org/taxon/urn:lsid:ipni.org:names:453031-1; or Lavandula officinalis https://powo.science.kew.org/taxon/urn:lsid:ipni.org:names:449067-1) This needs to be reviewed.

7.      “Official nomenclature of the International Code for the Nomenclature of Cultivated Plants (ICNCP)” This code is not used in the MS, as any cultivar is mentioned, only species.

8.      Cultural importance index. The use of this index can be justified when the dataset is big, and authors try to state the species with a higher cultural value in a study area. In this paper, the study area is very big (all the country, only based on 8 provinces and only 75 informants), and only focused on 1 family (/with only 9 wild species). Do authors believe that this would be approximatively for the cultural value of these taxa in the country? For criticism to the extensive (and bad) use of the index, see Leonti, 2022, J. Ethnopharm. 288, 115008).

9.      All the used indexes should be described with their formula.

Results

In general: The uses should be better described. Which are the medicinal uses of the included taxa? (condition, part of the plant used, administration, preparation, UR, etc.)

For 7 taxa, only one use was achieved. Is this a reflect of the low use of these lamiaceae, or may this reflect a need to perform more interviews or field research?

In a look to the tables, I cannot know which are the uses for the most cited taxa, M. spicata, which has 118 UR for 5 uses. –Which are them? How many UR for each? How is the RFC calculated? (add the formula)? This index is also confusing, as with this name, one can easily imagine that this shows that the 0.987 RFC for this species reflect that the 98% of citations were for this species (as the index is named “relative frequency of citation).   

Lavandula officinalis is not mentioned in the table. Moreover, are you sure the cultivated lavender is not L. x intermedia?

In table 2 we can see total UR= 423. This is a median of 423/79= 5.35 UR/informant

How can the CI for Mentha spicata be higher than 1? As you don’t provide the formula, I cannot check its calculation, but according to my knowledge, this index is formulated to achieve results in between 0 and 1. Please, explain.

Author Response

Dear Reviewer,

Thank you for the recommendations and for the careful review of our work. We are grateful for the detailed revisions aimed to improve the quality and readability of the current manuscript. We tried our best to correct the pointed-out shortages and to integrate your suggestions. Please, find bellow the responses to the specific comments.

The MS is generally well presented and written. The weak part is the used methods, and how authors show their results. In my view, the MS includes a very small amount of Lamiaceae species, considering the high biodiversity of wild ones (and more considering the variety of cultivated ones). Also, the interview effort seems to be weak (only 75 interviews for 29 taxa). Results show a low number of repetition of uses, and the extensive use of indexes is not justified, since the dataset is not big, and the somehow hide the low number of UR and studied taxa (or informants). What is the sense of describing in the table the RFC when it is just the UR/79?

Thank you for these comments. Considering the profile of the journal (Agronomy) and the extensive modern research on Lamiaceae as medicinal plants (popularized in numerous books, published since the 1950s in Bulgaria and also by a variety of media outlets and many amateur healers) we reckon that the cultivated taxa maintained in Bulgarian rural gardens and related local knowledge have received less attention and therefore we place the emphasis on that. As we mention in the discussion, the number of medicinal plants of Lamiaceae family used in the Bulgarian folk medicine was higher, reaching 45, according to the ethnographic sources (2013) [1]. However, there is some evidence that the knowledge is fading away and only 8 species were mentioned by “farmers, shepherds, local healers and old people” in the early 1990s [2]. The taxa used in recreational teas were reported to be 5 in 2013 [3] and no wild edible Lamiaceae taxa were mentioned by other researchers [4]. The discussion was changed so to highlight these changes.

Only few species were found commonly popular among most of the participants we questioned and the diversity of the taxa in the gardens was commensurable with the findings published in previous studies on home gardens, where, however, none of the Lamiaceae taxa was found present even in half of the studied gardens [5,6]. Given the number of the currently presented taxa (27) and the high occurrence frequency of some of them it will definitely be interesting to broaden the field research in more regions so to create more detailed picture of the utilization of Lamiaceae and to see if some species and/or traditional use(s) have remained undocumented. The participants are presented in detail in Table 1.

Weak points

abstract

  1. As the paper don't deal with chemical compounds or pharmacological properties, the sentence in abstract “ However, the extensive pharmacological research 12 throughout the years and universal commercialization of essential oils and extracts based 13 medicines derived from Lamiaceae plants have led to considerable replacement of the tradi-14 tional knowledge related to their use.” Is out of context

We considered your remark and removed the said sentence.

  1. “Current study aimed to document Lamiaceae taxa used…” / “Traditional and contemporary ways of utilization of Lamiaceae taxa as culinary 21 and medicinal plants, in herbal teas, as repellents, ritual plants, etc. are presented” in fact, as neither the table of results nor the sup. Mat. Deep on the real traditional use, describing the real use (use, medicinal use, condition, part used, administration / elaboration forms….) I would change this sentence.

Thank you for this remark. Traditional medicinal use of Lamiaceae taxa in Europe, incl. Bulgaria is reviewed by numerous authors. Our goal was not to repeat published and in some regards common information, but to show the multitude of their use as well as origin and sources of seeds and/or planting material as important factors for their conservation as valuable genetic resources. We haven't spotted new and unique medicinal use of Lamiacea taxa except anecdotal historical application of probably basil twigs as mechanical abortive and in some acupuncture/magical practices in the 1980s.

Methods

  1. Is the study covering all Bulgarian territory? At least all provinces or all the biogeographical different units?
  2. A map on the surveyed population would be appreciated, to know the extension of the study and how it can reflect the real situation in the country.

Thank you for this suggestion. A map was included as Figure 1. As it is obvious from the map we have studied gardens both in North and South Bulgaria which allows us to compare the diversity of Lamiaceae taxa on both sides of the Balkan range.

  1. Lamiaceae has 150 wild taxa in the Bulgarian flora. Is this paper with just 9 of them included, well designed?

We identified 27 taxa during our study, primarily focused on home gardens, which is about 60% of those known as medicinal plants in Bulgaria (wild and cultivated) [1]. Additional research would surely add more to this number; however, these additional taxa would be probably identifiable by more experienced healers or gatherers (also related to your next question) and undoubtedly would reveal more foreign species/cultivars. As we briefly mention in the end of the results “several wild Lamiaceae species (i.e. Ballota nigra L., Lamium amplexicaule L., L. maculatum L., L. purpureum L., Leunurus cardiaca L., Glechoma hederacea L., Salvia nemorosa L., S. pratensis L. S. verticillata L.)” were present in the home gardens of some of the participants. However, they were seen as weeds, regardless of the known traditional and modern use of some of them.

  1. Selection of informants. Ethnobotany use to deal with oral transmitted knowledge, as it study the folk knowledge, appart from the academic one. Then… your selection of informants is not good, as they were “was acquired so to identify prominent gardeners, agronomists and/or local healers, when needed.….” This is not ethnobotany.

Thank you for your remarks. Selection of the participants was following local people keenness to share information on the plants they cultivate and use. Bulgarians may not be classified as herbophobes but there is a certain preference to crop plants and personally grown plant food among Bulgarians which was recognized by ethnographers [7]. Naturally, people who are directly engaged in gardening/agriculture and/or have certain knowledge in folk medicine can provide more information on the subject. The said experts, born and living in villages and small towns, were clearly more capable to discern traditional local knowledge from academic/modern/foreign one (some of which was brought to the community exactly by them). Substantial depopulation and abandonment of rural areas in Bulgaria also contribute to lowering numbers of people who would be capable to supply such knowledge. Moreover, orally transmitted knowledge is dynamic, especially when cultivated plants are involved and very often prominent gardeners, agronomists and/or local healers are actually good “reference points” for recruitment of participants harboring local knowledge and/or preserving landraces/local forms.

  1. “Reference specimens and/or image data were collected for identification pur-112 poses; herbarium specimens were deposited in the Herbarium of the Institute of Bio-113 diversity and Ecosystem Research, Bulgarian Academy of Sciences (SOM).” Then, were are the voucher codes? They should be always added in the research (as mandatory in most journals dealing with the discipline)

Herbarium specimens were collected only when the garden owners allowed us to do so. We did not include images and/or numbers of all taxa as they are common and most of them in cultivation.

  1. Taxonomy and nomenclature. 1. Please make sure authorities are always added at least the first time each taxa is mentioned in the text (see, e.g., ). Some odf the used names, even being in consonance of ref 51, are not generally considered valid at the used range: e.g., Ocimum minimum (see, eg. https://powo.science.kew.org/taxon/urn:lsid:ipni.org:names:453031-1; or Lavandula officinalis https://powo.science.kew.org/taxon/urn:lsid:ipni.org:names:449067-1) This needs to be reviewed.

Thank you for these recommendations. We rechecked the text and the tables. Both Ocimum and Lavandula are introduced taxa for Bulgaria, as stated in table 2.

  1. “Official nomenclature of the International Code for the Nomenclature of Cultivated Plants (ICNCP)” This code is not used in the MS, as any cultivar is mentioned, only species.

Thank you for the remark, we removed the redundant information.

  1. Cultural importance index. The use of this index can be justified when the dataset is big, and authors try to state the species with a higher cultural value in a study area. In this paper, the study area is very big (all the country, only based on 8 provinces and only 75 informants), and only focused on 1 family (/with only 9 wild species). Do authors believe that this would be approximatively for the cultural value of these taxa in the country? For criticism to the extensive (and bad) use of the index, see Leonti, 2022, J. Ethnopharm. 288, 115008).
  2. All the used indexes should be described with their formula.

Thank you for pointing out that the generalizing use of the indices based on our sample could mislead further research in the frame of latest published literature. Data collected from more provinces would be more significant for such calculations and appropriate for further comparisons. The number of UR in different categories is given instead. We corrected table 2 and supplementary table 1 accordingly.

Results

In general: The uses should be better described. Which are the medicinal uses of the included taxa? (condition, part of the plant used, administration, preparation, UR, etc.)

Current paper focuses on the diverse uses of the documented Lamiaceae taxa and not on their (ethno)pharmacological relevance, a topic extensively discussed by many authors throughout the years. Some recent reviews could be found here:

https://www.mdpi.com/2223-7747/10/2/279

https://link.springer.com/article/10.1007/s11101-020-09690-9

https://www.mdpi.com/2223-7747/10/1/132

https://www.mdpi.com/1420-3049/26/12/3712

https://www.mdpi.com/journal/plants/special_issues/lamiaceae_species

https://www.sciencedirect.com/science/article/pii/S0026265X18312773

https://www.hindawi.com/journals/prm/2018/7801543/

https://www.frontiersin.org/articles/10.3389/fphar.2020.00852/full

For 7 taxa, only one use was achieved. Is this a reflect of the low use of these lamiaceae, or may this reflect a need to perform more interviews or field research?

Thank you for this question. The answer would depend on the taxon. For example Clinopodium dalmaticum L. has limited distribution in South Bulgaria and more interviews outside its distribution range would hardly increase the NU. As we can see from a recent ethnopharmacological research specifically targeting C. dalmaticum in Bulgaria the percentage of anecdotal reports is quite low – 13; it is known only as medicinal and aromatic plant used also for recreational teas [8]. Here we placed recreational teas in the medicinal use category.

In the case of Marrubium peregrinum higher number of interviews would probably reveal a medicinal use, replacing M. vulgare in folk remedies for respiratory and gastrointestinal disorders. However, all Bulgarian names of the species are derived from the Bulgarian word for broom which points to its most common use. The use of M. vulgare was expected to be more frequent, given it is known as medicinal plant in Bulgarian folk medicine, but the reports on its use were also few.

In a look to the tables, I cannot know which are the uses for the most cited taxa, M. spicata, which has 118 UR for 5 uses. –Which are them? How many UR for each? How is the RFC calculated? (add the formula)? This index is also confusing, as with this name, one can easily imagine that this shows that the 0.987 RFC for this species reflect that the 98% of citations were for this species (as the index is named “relative frequency of citation).

Thank you for pointing ways to improve the readability. We replaced the calculated indices with UR.

Lavandula officinalis is not mentioned in the table. Moreover, are you sure the cultivated lavender is not L. x intermedia?

Thank you for pointing our mistake in the text. The presented materials we identified as L. agustifolia.

In table 2 we can see total UR= 423. This is a median of 423/79= 5.35 UR/informant

Nearly 50% of the participants cultivated and/or used four and more Lamiaceae taxa.

How can the CI for Mentha spicata be higher than 1? As you don’t provide the formula, I cannot check its calculation, but according to my knowledge, this index is formulated to achieve results in between 0 and 1. Please, explain.

The maximum value of CI is the total number of different use-categories. [9]

Sincerely,

Ivanova & co-authors

  1. Georgiev, M. Bulgarian Folk Medicine; Vasileva, M., Georgiev, M., Georgieva, I., Penchev, V., Popov, R., Simeonova, G., Troeva, E., Tsaneva, E., Eds.; 2nd ed.; Prof. Marin Drinov Academic Publishing House: Sofia, 2013; ISBN 978-954-322-542-8.
  2. Ivancheva, S.; Stantcheva, B. Ethnobotanical inventory of medicinal plants in Bulgaria. J. Ethnopharmacol. 2000, 69, 165–172, doi:10.1016/S0378-8741(99)00129-4.
  3. Sõukand, R.; Quave, C.L.; Pieroni, A.; Pardo-de-Santayana, M.; Tardío, J.; Kalle, R.; Łuczaj, Ł.; Svanberg, I.; Kolosova, V.; Aceituno-Mata, L.; et al. Plants used for making recreational tea in Europe: A review based on specific research sites. J. Ethnobiol. Ethnomed. 2013, 9, 1–13, doi:10.1186/1746-4269-9-58/TABLES/4.
  4. Nedelcheva, A. An ethnobotanical study of wild edible plants in Bulgaria. EurAsian J. Biosci. 2013, 7, 77–94, doi:10.5053/EJOBIOS.2013.7.0.10.
  5. Vogl-Lukasser, B.; Vogl, C.R. The changing face of farmers’ home gardens: a diachronic analysis from Sillian (Eastern Tyrol, Austria). J. Ethnobiol. Ethnomedicine 2018 141 2018, 14, 1–20, doi:10.1186/S13002-018-0262-3.
  6. Reyes-García, V.; Aceituno, L.; Vila, S.; Calvet-Mir, L.; Garnatje, T.; Jesch, A.; Lastra, J.J.; Parada, M.; Rigat, M.; Vallès, J.; et al. Home Gardens in Three Mountain Regions of the Iberian Peninsula: Description, Motivation for Gardening, and Gross Financial Benefits. https://doi.org/10.1080/10440046.2011.627987 2012, 36, 249–270, doi:10.1080/10440046.2011.627987.
  7. Markova, M. Food and nutrition: between nature and culture.; Prof. Marin Drinov Academic Publishing House: Sofia, 2011; ISBN 978-954-322-462-3.
  8. Mincheva, I.; Jordanova, M.; Benbassat, N.; Aneva, I.; Kozuharova, E. Ethnobotany and exploitation of medicinal plants in the Rhodope Mountains – is there a hazard for Clinopodium dalmaticum? Pharmacia 2019, 66, 49–52, doi:10.3897/PHARMACIA.66.E35139.
  9. Tardío, J.; Pardo-De-Santayana, M. Cultural Importance Indices: A Comparative Analysis Based on the Useful Wild Plants of Southern Cantabria (Northern Spain)1. Econ. Bot. 2008, 62, 24–39, doi:10.1007/S12231-007-9004-5.

Reviewer 2 Report

This study evaluates the Lamiaceae taxa used in rural areas of Bulgaria and explores the related local knowledge and cultural practices that influence their utilization for various purposes.

The topic of paper is interesting and in the aim of the special issue of the journal. The paper requires minor revision because needs some additional details to be considered for publication.

Specific comments and suggestions for improving the paper are:

I suggest changing the title as: Traditional and modern uses instead of multipurpose utilization…..

Abstract, please, add the conclusion

Line 94, please, I suggest adding a map of Bulgaria regions

Line 288, Figura 4 is Mentha spicata…….please, check…

Author Response

Dear Reviewer,

Thank you for the positive review and for the careful consideration to our work. We are grateful for the detailed recommendations that will improve the quality and readability of the current manuscript. We tried our best to correct the pointed-out insufficiencies and to integrate your suggestions. Please, find bellow the responses to the specific comments.

I suggest changing the title as: Traditional and modern uses instead of multipurpose utilization…..

We propose the following change of the title: Lamiaceae plants in Bulgarian rural livelihoods – diversity, utilization and traditional knowledge

Abstract, please, add the conclusion

Thank you for this remark. We included the following: Recent knowledge on medicinal properties contributed to introduction of new taxa in the gardens (wild and cultivated), while traditional culinary practices were found to sustain the diversity of local forms (landraces).

Line 94, please, I suggest adding a map of Bulgaria regions

Thank you for the suggestion. Map of the studies regions is added as Figure 1.

Line 288, Figura 4 is Mentha spicata…….please, check…

We are obliged for this remark. We added suitable illustration  of Marrubium peregrinum broom (now Figure 6).

Sincerely,

Ivanova & co-authors

Reviewer 3 Report

Overall this is an interesting study, and I always feel it is important to document traditional knowledge.  However, the study sets up the threats that crop yields and quality face, however, I never felt the manuscript got back to that topic by the end of that article.  The authors describe Lamiaceae are well represented in the Bulgarian rural home gardens, but is that being threatened?  Are their factors that might cause these rural home gardens to disappear?  Does this study provide any information about if the value is being lost at the younger generations?  Is there evidence they are shifting to different types of home gardens?  These are several questions that I thought the study would address, but I don't think they were.  And I was left wondering what significant information does this study address, aside from the important documentation of rural plant gardens in Bulgaria which are understudied.  

I do not completely understand the purpose of the Jaccard Similarity index analysis; is there a benefit if they appear on multiple "lists" and are these lists just the different categories the plants are used for?  Yet, I was left wondering the point of the Jaccard similarity index.  

I was lefting wondering what this study means or what the bigger implications of this study were, and perhaps this manuscript could provide more information on why the findings are important.  The authors mention landraces, but how different are they in the gardens?  There we not studies done on those differences between them.  

Author Response

Dear Reviewer,

Thank you for the recommendations and for the careful review to our work. We are grateful for the detailed assessment that will improve the quality and readability of the current manuscript. We tried our best to correct the pointed-out insufficiencies and to integrate your suggestions. Please, find bellow the responses to the specific comments.

The authors describe Lamiaceae are well represented in the Bulgarian rural home gardens, but is that being threatened?  Are their factors that might cause these rural home gardens to disappear?  Does this study provide any information about if the value is being lost at the younger generations?  Is there evidence they are shifting to different types of home gardens?  These are several questions that I thought the study would address, but I don't think they were.  And I was left wondering what significant information does this study address, aside from the important documentation of rural plant gardens in Bulgaria which are understudied.

Thank you for your valuable remarks. A number of factors contribute to the abandonment of the local landraces and agrodiversity loss in general. In the current case gradual urbanization and growing depopulation and the ageing of the rural inhabitants threaten significantly the livelihoods in of these areas leading directly not only to the disappearance of the resources but also to the related knowledge. We rearranged and upgraded the discussion in this direction so to highlight the importance of multiple uses for the preservation of Lamiaceae resources.

I do not completely understand the purpose of the Jaccard Similarity index analysis; is there a benefit if they appear on multiple "lists" and are these lists just the different categories the plants are used for?  Yet, I was left wondering the point of the Jaccard similarity index. I was lefting wondering what this study means or what the bigger implications of this study were, and perhaps this manuscript could provide more information on why the findings are important. 

Thank you for these questions. As it is pointed out in the Results some of the participants utilize separate local forms (landraces) and purchased varieties of same species for different purposes. Jaccard SI is here by used to assess the diversity/similarity of the uses of Lamiaceae taxa based on the use-reports of the participants. Venn diagram only is a good overall illustration but the SI gives more precise value of the overlaps between the categories.

The authors mention landraces, but how different are they in the gardens?  There we not studies done on those differences between them.  

Thank you for this comment. Description and comparison (genetic and phytochemical) of the reported local forms (total of 140 from 4 species) deserves further and detailed study that would attract more interest to their conservation and/or introduction in larger scale production.

Sincerely,

Ivanova & co-authors

Round 2

Reviewer 1 Report

Dear authors. You are trying to publish a list of Lamiaceae with traditional uses. But at the end, the results just include: 

- 27 species, only 9 of them wild in Bulgaria (the checklist cites and as authors mention "nearly 150 taxa" wild in the country in the family. Apart, as they mention "is "the second most represented plant family in the Bulgarian rural home gardens". Among them, at least one should not be in the table: Ocimum minimum is just a cultivar of O. basilicum (check the taxonomical database I already mentioned in my past review). Nomenclature needs a further review (L. officinalis; and one of the most diverse genus, Thymus, is only mentioned as genus, without deeping on the taxonomical data and species which are used there (something really interesting). 

- Surveys in 34 settlements from 8 provinces. Yes, the map shows all the provinces, but not the few settlements. 

Only 75 interviews (so, very low number of repetitions and UR, with several uses with only 1 UR something unpublishable in most journals

- For 7 taxa, only one use was achieved

- Readers cannot trace the cultural use of each taxon: table 2 dont mention the use in each case, and it is not mentioned elsewhere, only some examples in the text. 

I think this is a weak research fieldwork to make the results of this research representative for the country (as the tittle indicates), or for a botanical family. Even being the second most used there, it is still one of the most important in the world. 

I suggest you to perform a better selection of informants, more interviews, dealing with a higher number of species. Then, perform the statistics, which as usual, are highly dependent on the total number of data. As it is now, tables 3 and 4 are badly indication anything (e.g., table 4 dont take into account species withing more than 1 category, and table 3 shows something striking: no correlation between culinary Lamiaceae and those for other purposes (medicinal!!!?), which of course, is due to the low number of uses for each species and a lack of review of the uses mentioned in other research for the country and species. 

Even recognising some merit for this MS, I see results are weak (also wrongly described) and further discusion is also weak. A different result would it be if you also have considered the previous literature (ref 1 you cited in the letter) and you analyse all data (both from the fieldwork of this research plus the literature review for Bulgaria and the species included). All results, statistics and conclusions would be different. 

Please, consider these words as just advices and real scientific criticism of your MS, based only in the results I can read and the discusion from these results. As an ethnobotanist with a quite long career, I see the fieldwork needs more work and data, a better descripcion of results (it is mandatory to add the voucher and description of uses!! (as I mentioned in my last review), apart from other data you already include as UR, or a proper citation of scientific names an authorities) .

Reviewer 3 Report

Updated discussion and change on the spin helps this manuscript significantly, however, the urgency of the article stresses the need to document these local forms, and yet it does not do genetic or phytochemical comparisons itself.  How different are the local forms?  Are some favored or sought out over others?  The authors do not describe if the local forms ranging from one family to another are maintained because of preference or convenience, or if these types of questions are asked during the interview.  Do the families take pride in having different forms?  Do other families try to obtain these other varieties, it's unclear how different the family forms are to me.  

Again, this is an interesting study, and the discussion helps stress the importance, however, I feel more needs to be done for this study to be complete.